# Free Radical Copolymerization of *N*-Isopropylacrylamide and 2,3-Dihydroxypropyl Methacrylate: Reaction Kinetics and Characterizations

**DOI:** 10.3390/ma18071614

**Published:** 2025-04-02

**Authors:** Zhishu Chen, Chao Zhang

**Affiliations:** 1School of Biomedical Engineering, Sun Yat-sen University, Shenzhen Campus, Shenzhen 518107, China; chenzhsh6@mail2.sysu.edu.cn; 2Guangdong Provincial Key Laboratory of Sensor Technology and Biomedical Instrument, Sun Yat-sen University, Shenzhen 518107, China

**Keywords:** poly(*N*-isopropylacrylamide), 2,3-dihydroxypropyl methacrylate, reaction kinetics, reactivity ratio

## Abstract

Poly(*N*-isopropylacrylamide) (PNIPAm) undergoes a sharp phase transition in aqueous solutions at around 32 °C, which is called the lower critical solution temperature; the tuning of the LCST of PNIPAm could be achieved by the copolymerization of *N*-isopropylacrylamide (NIPAm) with other hydrophilic/hydrophobic monomers to regulate the solvation state of PNIPAm and meet the requirements of possible applications. Herein, a hydrophilic monomer, 2,3-dihydroxypropyl methacrylate (DHPMA), w introduced to regulate the phase transition behavior of PNIPAm via free radical copolymerization. A series of poly(*N*-isopropylacrylamide-*co*-2,3-dihydroxypropyl methacrylate) (P(NIPAm-*co*-DHPMA)) was synthesized and characterized. The reaction kinetics were investigated in detail. In this copolymerization, the reactivity ratios of DHPMA and NIPAm were found to be 3.09 and 0.11, suggesting that DHPMA had greater preference for homopolymerization than for copolymerization, while NIPAm had greater preference for copolymerization than for homopolymerization. The phase transition temperature of P(NIPAm-*co*-DHPMA) copolymers varied from 31 to 42 °C by controlling the content of DHPMA in the copolymers from 0 to 58 mol%. Finally, the good cytocompatibility of P(NIPAm-*co*-DHPMA) was confirmed. These results provide insights into designing thermo-responsive polymers with suitable responsive behaviors that meet the requirements of different applications.

## 1. Introduction

Thermo-responsive polymers drastically change their physicochemical properties when the temperature reaches their phase transition temperatures, which has gained attention for applications in various fields [1]. Among these polymers, poly(*N*-isopropylacrylamide) (PNIPAm) is one of the most popular and classic thermo-responsive polymers, with 57 years of research history since the first systematic study of the phase diagram of PNIPAm was reported by Heskins and Guillet in 1968 [2]. Its lower critical solution temperature (LCST) had been identified to be approximately 32 °C in aqueous solutions [1,2,3], which is close to the physiological temperature, attracting interest in biomedical fields. The molecular chain of PNIPAm undergoes a coil-to-globule transition [4] driven by entropy changes in aqueous solutions in response to the increase in temperature, which transitions through the LCST [5,6]. These microscopic changes lead to the aggregation of PNIPAm-based materials and finally result in clear changes in the macroscopic properties of PNIPAm-based materials such as volume shrinkage [7], a decrease in light transmittance [8], conductivity changes [9], and a decrease in hydrophilicity [10]. Based on these fascinating properties, PNIPAm-based materials have been developed in biosensing [11,12], drug delivery [13,14], and tissue engineering [15,16].

The different requirements for thermo-responsive polymers in various applications pose a challenge to develop novel thermo-responsive polymers with a rational design. Copolymerization is a powerful strategy to build novel thermo-responsive polymers based on known thermo-responsive polymers with on-demand phase transition behavior for various applications [17,18,19,20]. Typically, hydrophilic comonomers increase PNIPAm solvation, raising the phase transition temperature. In contrast, hydrophobic comonomers reduce solvation, lowering the transition temperature [21]. However, the molecular interactions between comonomers also play an essential role in the phase transition behavior of the copolymer [5]. For example, 2-hydroxyethyl methacrylate (HEMA) is hydrophilic. However, it decreases the phase transition temperature of PNIPAm due to the hydrogen bond between the hydroxyl groups and carbonyl groups, which makes the copolymer more hydrophobic [22]. Zhou et al. investigated the micro-dynamic mechanism of the phase transition behavior of poly(*N*-isopropylacrylamide-*co*-2-hydroxyethyl methacrylate) hydrogels and found that the breakage of the hydrogen bond between the carbonyl groups of HEMA and water molecules as well as the breakage of hydrogen bonds between the carbonyl groups of HEMA and the amide groups of NIPAm at low temperatures created a hydrophobic environment in the hydrogel. This led to the breakage of the hydrogen bonds between the amide group of NIPAm and water molecules at lower temperatures and finally led to a lower phase transition temperature than PNIPAm [22]. The amide groups of NIPAm and the hydroxyl groups of HEMA then formed hydrogen bonds with each other [22]. Therefore, it is necessary to understand the regulation effect of comonomers on the phase transition behavior of PNIPAm to develop novel PNIPAm-based polymers on demand.

The reaction kinetics of copolymerization play an important role in optimizing the polymerization process to obtain the copolymer required [23]. Kinetic parameters such as the reactivity ratios of copolymerization are important to control the final structures of the copolymer [24,25]. Mathematical fitting methods such as the Fineman–Ross method [26], Tidwell–Mortimer method [27], and Kelen–Tüdös method [28], in combination with experimental kinetic data, are used to determine the reactivity ratio.

2,3-Dihydroxypropyl methacrylate (DHPMA), otherwise known as glycerol monomethacrylate [29], is a promising monomer in biomedical fields that has been used in commercialized contact lenses [30,31] and has many studies in biosensing [32], cell culture [33], and tissue engineering [34]. Its homopolymers have been recognized as a potential candidate of poly(ethylene glycol) in biomedical fields because of its hydrophilic, biocompatible, non-toxic, and potential stealth effect [25,33,35]. In contrast to HEMA, DHPMA has one more hydroxyl group and has been found to increase the phase transition temperature higher than PNIPAm after the reversible addition-fragmentation chain transfer (RAFT) copolymerization of DHPMA and NIPAm [25]. Saldívar-Guerra et al. investigated the reactivity ratios of DHPMA and other comonomers, including NIPAm, and found that the reactivity ratio of DHPMA was 2.55, while the reactivity ratio of NIPAm was 0.11 [25]. However, the reactivity ratios of DHPMA and NIPAm remain to be confirmed due to the existence of 22 mol% of a DHPMA isomer, 1,3-dihydroxypropyl methacrylate, in the previous report. The reaction kinetics of the free radical copolymerization of NIPAm and DHPMA remained unexplored.

In this study, a series of poly(*N*-isopropylacrylamide-*co*-2,3-dihydroxypropyl methacrylate) (P(NIPAm-*co*-DHPMA)) was synthesized via free radical copolymerization. The chemical structure of P(NIPAm-*co*-DHPMA) was carefully confirmed by nuclear magnetic resonance (NMR) spectroscopy. The reaction kinetics of the copolymerization were investigated using ^1^H nuclear magnetic resonance (^1^H NMR) spectra and elemental analyses. The phase transition behavior of P(NIPAm-*co*-DHPMA) in an aqueous solution was evaluated. As the results show, a series of poly(*N*-isopropylacrylamide-*co*-2,3-dihydroxypropyl methacrylate) with different contents of NIPAm and DHPMA units were successfully synthesized. The conversion and the reactivity ratios of the comonomers were produced. These findings provide a design for P(NIPAm-*co*-DHPMA) copolymers with precisely controlled phase transition behavior matching different criteria.

## 2. Materials and Methods

### 2.1. Materials

2,3-Dihydroxypropyl methacrylate (DHPMA) (95%; Beijing HwrkChemical Technology Co., Ltd., Beijing, China) was further purified using vacuum distillation to ensure a purity above 99% (confirmed by ^1^H NMR) before use. *N*-isopropylacrylamide (NIPAm) (98%; Macklin, Shanghai, China) was recrystallized from *n*-Hexane (≥99.7%; Guangzhou Chemical Reagent Factory, Guangzhou, China)/methylbenzene (≥99.7%; Guangzhou Chemical Reagent Factory, Guangzhou, China) (50/50, *v*/*v*) via dissolution at 40 °C and then cooled under refrigeration. 2,2′-Azobis(2-methylpropionitrile) (AIBN) (98%; Macklin, Shanghai, China) was recrystallized from ethanol (≥99.7%; Guangzhou Chemical Reagent Factory, Guangzhou, Guangdong, China) via dissolution at 50 °C and then cooled under refrigeration. Anhydrous dimethyl sulfoxide (DMSO) (99.7%; Macklin, Shanghai, China), copper(I) chloride (CuCl) (99%; Bidepharm, Shanghai, China), 1,4-dinitrobenzene (98%; Sigma-Aldrich, Darmstadt, Germany), and deuterated dimethyl sulfoxide (DMSO-*d*_6_) (99.9% with 0.03% TMS; Cambridge Isotope Laboratories, Inc., Tewksbury, MA, USA) were used as received. The poly(methacrylic acid methyl ester) (PMMA) standard and *N*,*N*-dimethylformamide (DMF; 99.8%) were products of Aladdin, Shanghai, China.

NIH 3T3 cells were a gift from the Hospital of Stomatology, Sun Yat-sen University. High-glucose Dulbecco’s Modified Eagle Medium (DMEM; Gibco Life Technologies, Inc., Grand Island, NY, USA) supplemented with 10% fetal bovine serum (FBS; Gibco Life Technologies, Inc., Grand Island, NY, USA) and 1% penicillin–streptomycin (Gibco Life Technologies, Inc., Grand Island, NY, USA) was used as the culture medium for NIH 3T3 cells. A Cell-Counting Kit-8 (CCK-8) was obtained from Beyotime Biotechnology, Shanghai, China. A Live/Dead Viability/Cytotoxicity Assay Kit for Animal Cells was obtained from KeyGEN BioTECH, Nanjing, Jiangsu, China.

### 2.2. Synthesis of P(NIPAm-co-DHPMA)

P(NIPAm-*co*-DHPMA) was synthesized via free radical copolymerization. In a typical reaction, 33.0 mmol of comonomers in total and 0.5 mol% AIBN (Appendix A) were dissolved in 30.0 mL DMSO in a Schlenk flask. The solution was first purged with nitrogen (99.99%) for 30 min, followed by three vacuum/nitrogen purge cycles. Then, the flask was immersed in a 70 °C water bath with magnetic stirring. At predetermined time points (set as 0, 10, 20, 30, 60, and 120 min), 200 μL of the reaction mixture was sampled and then mixed with 400.0 μL DMSO-*d*_6_ containing 72.0 μg CuCl and 3.0 mg 1,4-dinitrobenzene. The mixtures were subjected to NMR measurements to determine the conversion of monomers at certain time points. After 2 h, the reaction was quenched in ice water and the reaction mixture was dialyzed against deionized (DI) water in a dialysis tube (molecular weight cut-off (MWCO) of 1000 Da) for 72 h with frequent changes of DI water. Finally, the dialysate was lyophilized before characterization.

### 2.3. Structural Characterization

The chemical structure of the copolymerization product was investigated using NMR spectroscopy. ^1^H NMR spectra, ^13^C nuclear magnetic resonance (^13^C NMR) spectra, distortionless enhancement by polarization transfer (DEPT; θ = 135°) spectra, ^1^H-^1^H correlation (^1^H-^1^H COSY) spectra, and ^1^H-^13^C heteronuclear single quantum coherence (HSQC) spectra were recorded using a Bruker Advance 400 spectrometer (Bruker, Billerica, MA, USA). All NMR measurements were conducted using DMSO-*d*_6_ as the solvent and TMS as the internal standard. The experimental settings of ^1^H NMR, ^13^C NMR, DEPT, ^1^H-^1^H COSY, and HSQC were as follows:

For ^1^H NMR, the pulse sequence, relaxation delay, number of scans, and spectral size were set as zg30, 1.00 s, 16, and 64 K, respectively. For ^13^C NMR, the pulse sequence, relaxation delay, number of scans, and spectral size were set as zgpg30, 1.55 s, 3600, and 64 K, respectively. For DEPT, the pulse sequence, relaxation delay, number of scans, and spectral size were set as dept135, 1.00 s, 2048, and 64 K, respectively. For ^1^H-^1^H COSY, the pulse sequence, relaxation delay, number of scans, and spectral size were set as cosygpdf, 1.00 s, 16, and (1024, 1024), respectively. For HSQC, the pulse sequence, relaxation delay, number of scans, and spectral size were set as hsqcedetgp, 0.9050 s, 16, and (1024, 1024), respectively.

### 2.4. Molecular Molar Mass

The molecular molar mass and polydispersity index (PDI) of the copolymerization products were determined using gel permeation chromatography (GPC) on an Agilent 1260 high-performance liquid chromatography system (Agilent Technologies, Inc., Santa Clara, CA, USA) with a Plgel 5 μm MIXED-C column (Agilent Technologies, Inc., Santa Clara, CA, USA). DMF was selected as the eluent with a flow of 1.0 mL/min at 35 °C. A light-scattering detector, a differential refractive index detector, and a viscometer were used to estimate the molar mass of the product and the results were calibrated using PMMA standards. The synthesized products were dissolved in DMF at a concentration of 1.0 mg/mL and filtered through a 0.22 μm membrane for each GPC characterization.

### 2.5. Conversion of Monomers

The conversion of the monomers was determined using ^1^H NMR spectroscopy. Before testing, the longitudinal relaxation times (T_1_) of NIPAm (δ = 5.54 ppm) and DHPMA (δ = 5.67 ppm) were measured following the inversion recovery method using nonlinear fitting with the following formula:(1)$$I(\tau )=I(0)+Pexp(-\frac{\tau }{T_{1}})$$
where τ is the relaxation time, *I*(*τ*) is the integral area of the peaks after a relaxation time of *τ*, P is a constant, and T_1_ is the longitudinal relaxation time.

The T_1_ of 1,4-dinitrobenzene (δ = 8.46 ppm) from the literature was used in this study [36]. The value of the relaxation delay (D_1_) was decided by using the largest T_1_ of NIPAm (δ = 5.54 ppm), DHPMA (δ = 5.67 ppm) (Appendix A), and 1,4-dinitrobenzene (δ = 8.46 ppm).

For ^1^H NMR spectroscopy, the pulse sequence, D_1_, number of scans, and spectral size were set as zg, 30 s, 32, and 64 K, respectively.

1,4-dinitrobenzene was set as the internal standard compound and its characteristic peak at 8.46 ppm was normalized as 1.00. The conversions of NIPAm and DHPMA were calculated using the normalized peak areas of NIPAm (δ = 5.54 ppm) and DHPMA (δ = 5.67 ppm) at different time points. The conversion of the monomers was calculated using the following formulae:(2)$$\alpha_{NIPAm,x}=\frac{A_{NIPAm,0}-A_{NIPAm,x}}{A_{NIPAm,0}}\times 100\%$$(3)$$\alpha_{DHPMA,x}=\frac{A_{DHPMA,0}-A_{DHPMA,x}}{A_{DHPMA,0}}\times 100\%$$(4)$$\alpha_{Total,x}=\frac{A_{DHPMA,0}+A_{NIPAm,0}-A_{NIPAm,x}-A_{DHPMA,x}}{A_{DHPMA,0}+A_{NIPAm,0}}\times 100\%$$

Here, α_NIPAm,*x*_, α_DHPMA,*x*_, and α_Total,*x*_ are the conversion of NIPAm, DHPMA, and total monomers at *x* min, respectively. *A*_NIPAm,0_ and *A*_DHPMA,0_ are the normalized peak areas of NIPAm (δ = 5.54 ppm) and DHPMA (δ = 5.67 ppm) at 0 min, while *A*_NIPAm,*x*_ and *A*_DHPMA,*x*_ are the normalized peak areas of NIPAm (δ = 5.54 ppm) and DHPMA (δ = 5.67 ppm) at *x* min.

### 2.6. Composition of the Copolymer

The C/H/N and O contents in the copolymer were determined using a Vario EL Cube (Elementar, Frankfurt, Germany) elemental analyzer and a Vario EL Cube CHNS/O (Elementar, Frankfurt, Germany) elemental analyzer, respectively; each type of copolymer was tested twice and the results were averaged (Appendix A).

The molar ratio (*F*) of the DHPMA unit (d[M_1_]) and NIPAm unit (d[M_2_]) in the copolymer were calculated using the following formula:(5)$$F=d\left[M_{1}\right]/d\left[M_{2}\right]=[(\left[O\right]/16.00)/(\left[N\right]/14.01)-1]/4$$

Here, [O] and [N] are the mass fractions of elements O and N in the copolymer, respectively.

### 2.7. Reactivity Ratios

After an 8 min reaction, samples were collected and quenched immediately to terminate the polymerization and calculate the reactivity ratios. The conversions of monomers were quantified to be less than 10% using the ^1^H NMR method. The composition of the copolymer was calculated using the elemental analysis results (Appendix A). The reactivity ratios of NIPAm and DHPMA were calculated using the Fineman–Ross method [26]. The concentration ratio (*f*) of DHPMA ([M_1_]) and NIPAm ([M_2_]) was ascertained as follows:(6)$$f=\frac{A_{DHPMA,x}}{A_{NIPAm,x}}$$

The molar ratio (*F*) of the DHPMA unit (d[M_1_]) and NIPAm unit (d[M_2_]) in the copolymer was calculated using the content of element N and element O obtained from the elemental analysis.

The reactivity ratios of DHPMA (*r*_1_) and NIPAm (*r*_2_) were obtained by a linear fitting with the following equation:(7)$$f(F-1)/F=f^{2}/F\times r_{1}-r_{2}$$

Sequence length distributions of NIPAm and DHPMA in the copolymer were calculated using the following formulae:(8)$$\underline{N}{\left(DHPMA\right)}_{x}={\left(\frac{r_{1}}{r_{1}+1/A}\right)}^{x-1}(\frac{1}{r_{1}A+1})$$(9)$$\underline{N}{\left(NIPAm\right)}_{x}={\left(\frac{r_{2}}{r_{2}+A}\right)}^{x-1}(\frac{1}{r_{2}/A+1})$$
where *x* represents the sequence length of the *x*DHPMA sequence (NIPAm-DHPMA*_x_*-NIPAm) or *x*NIPAm sequence (DHPMA-NIPAm*_x_*-DHPMA) in the copolymer and *N*(DHPMA)*_x_* and *N*(NIPAm)*_x_* represent the percentage of each sequence length in the DHPMA sequence and NIPAm sequence, respectively. *A* represents the theoretical feeding ratios of DHPMA and NIPAm ([DHPMA]/[NIPAm]).

### 2.8. Cloud Point

To determine the phase transition temperature of the copolymer, the transmittance of 0.2 wt% P(NIPAm-*co*-DHPMA) in deionized water was recorded at 500 nm in the temperature range of 25 °C to 50 °C with a step of 0.5 °C using a 721 Visible Spectrometer (Tianjin Taisite Analytical Instrument Co., Ltd., Tianjin, China). The cloud point (T_cp_) was defined as the temperature corresponding with the midpoint of the transmittance. The phase transition range was defined as the temperature region where the transmittance decreased. Triplicate samples were measured for each step.

### 2.9. In Vitro Cytotoxicity

To assess the cytotoxicity of P(NIPAm-*co*-DHPMA), 5 × 10^3^ NIH 3T3 cells in 100 μL of the culture medium were seeded in each well of a 96-well plate and incubated at 37 °C in a 5% CO_2_ atmosphere for 24 h; then, the culture medium was replaced by 100 μL of a fresh medium containing P(NIPAm-*co*-DHPMA) at various concentrations (10 mg/mL and 0 mg/mL (Control group)) and the cells were incubated for another 24 h. Subsequently, the culture medium was replaced by 110 μL of a fresh medium containing 10 μL of the CCK-8 solution and incubated for an additional 1 h, then the absorbance of each well at 450 nm was measured using an AMR-100 microplate reader (Allsheng Instruments Co., Ltd., Hangzhou, China); five duplicates were measured for each group. The cell viability was calculated according to the following formula:(10)$$Cell\;viability\%=(A_{sample}-A_{background})/(A_{control}-A_{background})\times 100$$

The data were then analyzed via a one-way ANOVA with Tukey’s post hoc test using OriginPro 2024. The difference between each group was considered to be statistically significant at *p* < 0.05.

Calcein-AM/PI staining was used to observe the live and dead cells after co-culture with P(NIPAm-*co*-DHPMA). Briefly, NIH 3T3 cells were seeded in a 24-well plate at a density of 5 × 10^4^ cells per well. The cells were incubated in 1.0 mL of the culture medium in a 5% CO_2_ atmosphere at 37 °C for 24 h. Then, the culture medium was replaced by 1.0 mL of fresh medium containing P(NIPAm-*co*-DHPMA) (10 mg/mL and 0 mg/mL (Control group)). The cells were incubated for another 24 h. Subsequently, the cells were stained and observed under an Olympus IX71 fluorescence microscope (Olympus, Tokyo, Japan).

## 3. Results and Discussion

### 3.1. Chemical Structure of the Copolymers

Before copolymerization, DHPMA was vacuum-distilled and the structure was confirmed by ^1^H NMR (Appendix A). Generally, DHPMA contains minor isomers, 1,3-dihydroxypropyl methacrylate, because the isomerization reaction of DHPMA continues to occur during storage, which involves the migration of the ester group [37,38]. The equilibrium levels of DHPMA and 1,3-dihydroxypropyl methacrylate have been reported to be 90% and 10%, even as monomer units in the polymer [37,38].

A series of P(NIPAm-*co*-DHPMA) copolymers with different feeding ratios were synthesized using free radical copolymerization and were named PND 8-2, PND 7-3, and PND 6-4, respectively. Their chemical structures were confirmed by ^1^H NMR and ^13^C NMR spectra (Figure 1a,b and Appendix A). The attribution of the peaks in the ^1^H NMR and ^13^C NMR spectra was evidenced by DEPT, ^1^H-^1^H COSY, and HSQC spectra (Figure 1c–e; Appendix A) and are summarized in Table 1.

The carbon atom of methyne (f’) was located at 76.37 ppm in ^13^C NMR, while the proton of methyne (f’) was located in the 4.61–4.83 ppm region in ^1^H NMR and was covered by the signal of other peaks [40]. The peaks of the 1,3-dihydroxypropyl methacrylate units in the copolymers were weak compared with the NIPAm and DHPMA units, indicating that the main chemical structure of P(NIPAm-*co*-DHPMA) comprised NIPAm units and DHPMA units.

The molecular molar mass and distribution of synthesized copolymers were determined using GPC measurements (Appendix A). The number average molar mass (M_n_) and PDI were dependent on the feeding ratio of DHPMA/NIPAm (Appendix A). When the feeding ratio increased from 2/8 to 3/7 (DHPMA/NIPAm), M_n_ increased from 57,820 g/mol to 234,160 g/mol, while PDI decreased from 3.79 to 1.82. Further increasing the feed ratio to 4/6 led to a decrease in both M_n_ (217,880 g/mol) and PDI (1.66).

### 3.2. Kinetics of Copolymerization

The kinetics of copolymerization were monitored using ^1^H NMR spectroscopy and elemental analyses. In this work, DHPMA contained 10% 1,3-dihydroxypropyl methacrylate after purification. The investigation of the copolymerization of NIPAm and DHPMA at this DHPMA purity was suitable for guiding the industrial production of materials. Based on the T_1_ value of DHPMA (δ = 5.67 ppm), NIPAm (δ = 5.54 ppm) (Appendix A), and 1,4-dinitrobenzene (δ = 8.46 ppm), the value of D_1_ = 30 s was used to obtain 99% of the equilibrium magnetization [41].

The conversion of monomers was calculated based on the integral of the peaks of the characteristic protons of monomers in the ^1^H NMR spectra at different copolymerization times (Figure 2a–c and Appendix A). After a 2 h reaction, the conversion of DHPMA reached over 90%, while the conversion of NIPAm was over 60%. The consumption of DHPMA was much faster than NIPAm, suggesting that the DHPMA content in the formed copolymers was higher at the beginning of copolymerization and decreased as the reaction progressed. The total conversion of the comonomers decreased from 94% to 77% when the DHPMA/NIPAm feeding ratio increased from 2/8 to 4/6, suggesting that the reaction rate of copolymerization decreased with an increase in the DHPMA/NIPAm feeding ratio. At a DHPMA/NIPAm feed ratio of 2/8, the conversion of DHPMA reached 100%; that of NIPAm reached approx. 60% after one hour’s reaction, suggesting the homopolymerization of NIPAm happened after the complete consumption of DHPMA.

A Fineman–Ross plot was harnessed to derive the reactivity ratios of the comonomers based on the conversion of monomers and the composition of products after an 8 min copolymerization (Figure 2d and Appendix A). All the conversions of the comonomers were confirmed to be no more than 10% after 8 min of copolymerization by ^1^H NMR. The reactivity ratio of DHPMA was 3.09, suggesting that DHPMA had greater preference for homopolymerization than for copolymerization. The reactivity ratio of NIPAm was 0.11, indicating that NIPAm had greater preference for copolymerization than for homopolymerization. DHPMA has previously been reported to contain minor isomers, 1,3-dihydroxypropyl methacrylate. The reactivity ratio of DHPMA obtained in this work was higher than the value of 2.55 previously reported [25], based on unpurified DHPMA (purity of only 90%). The reactivity ratios of DHPMA and 1,3-dihydroxypropyl methacrylate have been reported to be 0.7 and 1.6, respectively [42]. Therefore, the different values of the reactivity ratio of DHPMA could be attributed to either the lower purity of the starting materials or the actually lower content of 1,3-dihydroxypropyl methacrylate in this work. These results suggest that DHPMA with a lower minor isomer ratio had a higher preference for homopolymerization during the copolymerization of NIPAm and DHPMA.

Based on the reactivity ratios, one could calculate the instantaneous sequences for a copolymer system [25] (Figure 3). The most abundant DHPMA sequence was the NIPAm-DHPMA-NIPAm sequence (Figure 3a). The percentage of the NIPAm-DHPMA-NIPAm sequence decreased from 56.42% to 32.68% with an increase in the DHPMA/NIPAm feeding ratio from 2/8 to 4/6, while the percentage of other DHPMA sequences increased. The most abundant NIPAm sequence was the DHPMA-NIPAm-DHPMA sequence (Figure 3b). The percentage of the DHPMA-NIPAm-DHPMA sequence increased from 69.44% to 85.84% with an increase in the DHPMA/NIPAm feeding ratio from 2/8 to 4/6, while the percentage of other NIPAm sequences decreased.

### 3.3. Phase Transition Behavior

The phase transition behavior of the P(NIPAm-*co*-DHPMA) aqueous solution was investigated at a macroscopic level using the cloud point method. The P(NIPAm-*co*-DHPMA) aqueous solution turned from clear to turbid upon increasing the temperature to T_cp_, exhibiting characteristic LCST behavior (Figure 4). The T_cp_ of P(NIPAm-*co*-DHPMA) increased from 33.0 °C to 42.1 °C, with the DHPMA content increasing from 34.21 mol% to 57.98 mol% (Table 2), and was higher than PNIPAm (31.1 °C; this work). The molecular structure of a DHPMA unit comprises two hydroxyl groups and one ester group, which can form more hydrogen bonds compared with NIPAm. Therefore, P(NIPAm-*co*-DHPMA) could form more hydrogen bonds compared with PNIPAm. From the perspective of thermodynamics, the increase in hydrogen bonds between the polymers increased the value of negative mixing enthalpy (|ΔH_mix_|) and then increased the phase transition temperature, which turned the Gibbs energy of mixing (ΔG_mix_) from negative to positive [5].

Compared with reversible addition-fragmentation chain transfer copolymerization [25], the copolymer synthesized using free radical polymerization had a higher PDI because of uncontrollable polymerization. Therefore, the products of free radical copolymerization in this work comprised a series of molecular chains with different DHPMA contents, which finally led to a wider phase transition range [43]. These findings provide a one-step method to fabricate a thermo-responsive polymer for a specific temperature range, which has potential application in biomedical fields. For example, the photothermal treatment temperature can be controlled by the transmittance of a temperature response coverage device [44].

### 3.4. In Vitro Cytotoxicity

To evaluate the cytocompatibility of P(NIPAm-*co*-DHPMA), NIH 3T3 cells were co-cultured with the copolymer for 24 h and stained. Almost no dead cells were detected at concentrations up to 10 mg/mL, and no difference in the cell morphology was observed compared with the Control group (Figure 5a–c). The cell viability in the presence of P(NIPAm-*co*-DHPMA) was above 80% for all studied copolymers and showed no significant difference between the Control group and experimental groups (Figure 5d). These results suggested the good cytocompatibility of the synthesized P(NIPAm-*co*-DHPMA).

## 4. Conclusions

In this work, DHPMA with 10% isomers was used to regulate the phase transition behavior of PNIPAm. A series of thermo-responsive copolymers (P(NIPAm-*co*-DHPMA)) were synthesized via free radical copolymerization. Characterizations of the copolymerization process and products were carried out, and the following findings were obtained:The reaction kinetics of the free radical copolymerization of NIPAm and DHPMA were investigated. After two hours’ copolymerization, the conversion of DHPMA and NIPAm reached at least 90% and 60%, respectively. The reactivity ratios of DHPMA and NIPAm were 3.09 and 0.11, respectively. DHPMA with a lower isomer content had a higher preference for homopolymerization.The NIPAm-DHPMA-NIPAm and the DHPMA-NIPAm-DHPMA sequences were the most abundant in the copolymer, suggesting a preferred random distribution of the two structural units in the main chain of the copolymer.The cloud point of P(NIPAm-*co*-DHPMA) increased from 31 to 42 °C with an increase in the content of DHPMA from 0 to 58 mol%; these copolymers exhibited a wide phase transition temperature range.Finally, the good cytocompatibility of P(NIPAm-*co*-DHPMA) copolymers was confirmed.

## Figures and Tables

**Figure 1 materials-18-01614-f001:**
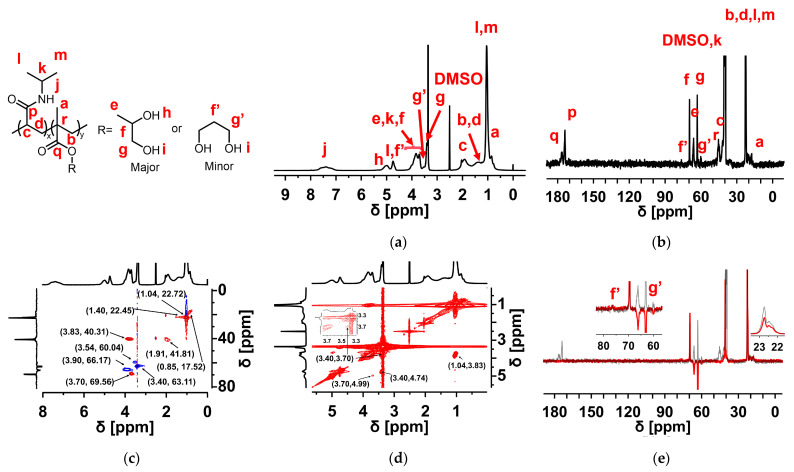
Structure of PND 7-3 copolymer: (**a**) ^1^H NMR, (**b**) ^13^C NMR, (**c**) HSQC, (**d**) ^1^H-^1^H COSY, and (**e**) DEPT spectrum (135°; red) vs. ^13^C NMR (gray).

**Figure 2 materials-18-01614-f002:**
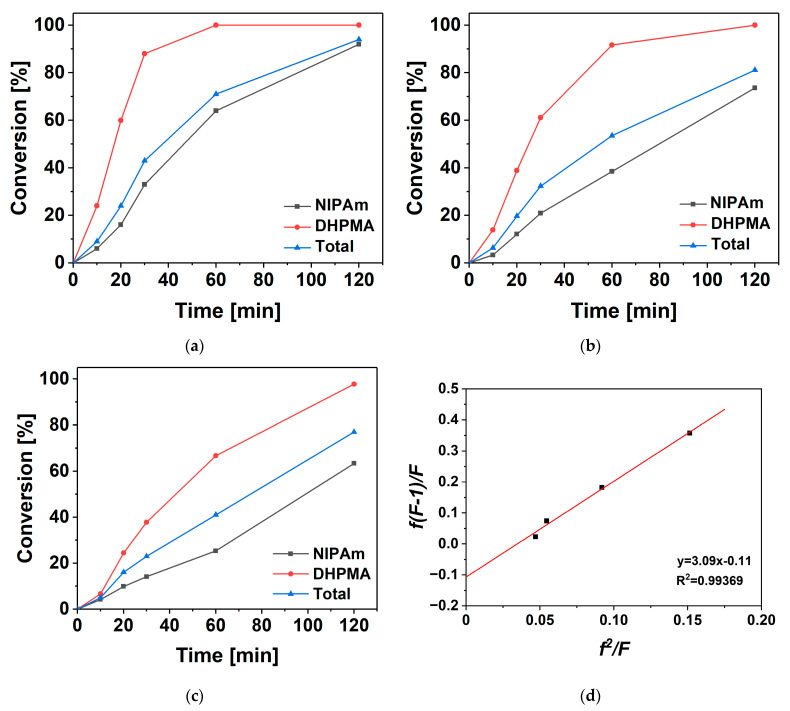
Kinetics of copolymerization. Conversion of (**a**) PND 8-2, (**b**) PND 7-3, and (**c**) PND 6-4 groups. (**d**) Fineman–Ross plot.

**Figure 3 materials-18-01614-f003:**
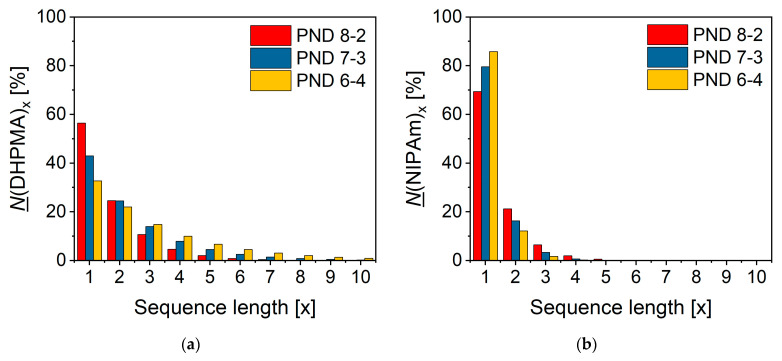
Sequence length distribution for the copolymerization of NIPAm and DHPMA at different feeding ratios. The vertical axis represents the percentage of the sequence length for (**a**) DHPMA and (**b**) NIPAm. A sequence length longer than 10 existed, but the percentage of these sequences was less than 2%.

**Figure 4 materials-18-01614-f004:**
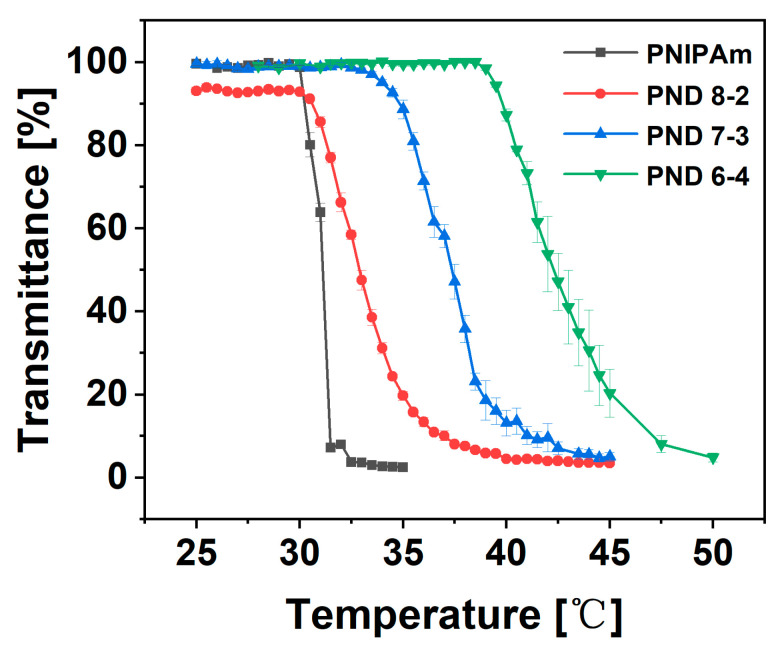
Phase transition behavior of P(NIPAm-*co*-DHPMA).

**Figure 5 materials-18-01614-f005:**
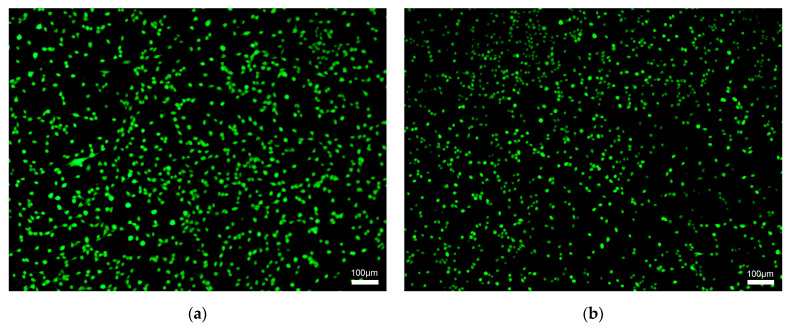
In vitro cytotoxicity of P(NIPAm-*co*-DHPMA). Live/Dead staining of (**a**) Control group, (**b**) PND 7-3, and (**c**) PND 6-4. Scale bar: 100 μm. (**d**) Cell viability. * *p* < 0.05; NS: no significance.

**Table 1 materials-18-01614-t001:** Chemical shifts and their attribution from the ^1^H NMR and ^13^C NMR spectra of the PND 7-3 copolymer.

^1^H NMR		^13^C NMR	
Chemical Shift [ppm]	Attribution	Chemical Shift [ppm]	Attribution
7.38	1H, –NH– [39]	176.44	C=O_DHPMA_
4.99	1H, –CH_2_–CH–OH	173.88	C=O_NIPAm_
4.74	1H, –CH–CH_2_–OH	69.56	–CH_2_–CH–OH
3.90	2H, –O–CH_2_–	66.17	–O–CH_2_–
3.83	1H, –CH–(CH_3_)_2_	63.11	–CH–CH_2_–OH
3.70	1H, –CH_2_–CH–OH	60.04	–CH(CH_2_OH)_2_
3.54	–CH(CH_2_OH)_2_	45.33	–C(CH_3_)–CH_2_–
3.40	2H, –CH–CH_2_–OH	41.81	–CH–CH_2_–
1.91	1H, –CH–CH_2_–	40.31	–CH–(CH_3_)_2_
1.40	2H, –CH–CH_2_–	22.72	–CH–(CH_3_)_2_
1.04	3H, –CH–(CH_3_)_2_	22.45	–CH–CH_2_–
0.85	3H, –C–CH_3_	17.52	–C–CH_3_

**Table 2 materials-18-01614-t002:** The T_cp_ and phase transition range of the copolymers.

Polymer Name	Feeding Ratio ([M_1_]/[M_2_])	DHPMA Content [mol%]	Phase Transition Range [°C]	T_cp_ [°C]
PNIPAm	0	0	30.0–32.5	31.1
PND 8-2	0.25	34.21	30.0–40.0	33.0
PND 7-3	0.43	45.65	32.0–43.5	36.5
PND 6-4	0.67	57.98	38.5–50.0	42.1

## Data Availability

The original contributions presented in this study are included in the article/Appendix A. Further inquiries can be directed to the corresponding author.

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
