# Peer review of "Free Radical Copolymerization of *N*-Isopropylacrylamide and 2,3-Dihydroxypropyl Methacrylate: Reaction Kinetics and Characterizations"

_materials, 2025, doi:10.3390/ma18071614_

Round 1

Reviewer 1 Report

Comments and Suggestions for Authors

This paper studies the "Free Radical Copolymerization of N-isopropylacrylamide and 2,3-dihydroxypropyl Methacrylate: Reaction Kinetics and Characterizations" which is an interesting topic, while there are still some problems that need to be addressed as follows:
1. There are some grammar mistake that need to be revised, please ask a native English writer help to review the whole paper, for example Typically, hydrophilicity comonomers increase the solvation of PNIPAm and lead to a higher phase transition temperature than PNIPAm, while hydrophobicity comonomers weaken the solvation and lead to a lower phase transition temperature."
can be "Hydrophilic comonomers increase PNIPAm solvation, raising the phase transition temperature. In contrast, hydrophobic comonomers reduce solvation, lowering the transition temperature"
2. Some figures lack of error bar, please add if possible. and how many samples are tested for each test, please also explain.
3. The abstract lacks specific value to support the conclusion
4. Additional details on statistical significance and analysis will help to better explain the discussion section.
5. More scientific discussion is recommended in the results discussion section.
6. Please explain the innovation of this study and the need for this study.
7. Some reference is far old like 1968 paper. Please consider adding a more recent study.
8. How to understand the trend in Figure 3. Please explain.
9. The conclusion can be more readable, Please revise it.

10.  some format of the reference is not right. please follow the journal requirement. 

Comments on the Quality of English Language

The English could be improved to more clearly express the research.

Reviewer 2 Report

Comments and Suggestions for Authors

I have the following comments on the submitted article:
1. I ask the authors to reduce the value of the agreement, which is at the level of 24%.
2. Please add Table S1-S6 and Figure S1-S8, which you mention in the content of the article.
3. Figures 1, 2, 3, 4 and 5 - edit the image to make it more transparent, label a-e in the images more clearly, separate the individual images a-e more from each other, because they merge too much and it is not clear which text belongs to which. Does the image on the top right belong to Figure 1b? Also edit/insert the designations of quantities in square brackets, e.g. time [min] etc. Add removable scales to Figures 5abc.
4. Table 1 - Also edit/insert the designations of quantities in square brackets.
5. In the introduction, list the citations, or use names for specific citations, e.g. Author et al. investigated ..... and found that...
6. Write the conclusion in points according to the individual findings.
7. Write lines 243 - 250 in a clear table.
8. Use the same spacing between the image, text and image caption (Fig. 3).

Reviewer 3 Report

Comments and Suggestions for Authors

The authors of the manuscript entitled "Free radical copolymerization of N-isopropylacrylamide and 2,3-dihydroxypropyl methacrylate: Reaction kinetics and characterizations" have discussed the properties of the (P(NIPAm-co-DHPMA)), a copolymer obtained by N-isoproylacrylamide  and 2,3-dihydroxypropyl methacrylate. Attention of the authors was focused about the increase in the LCST according to the change in the ratio of the two co-monomers. Characterization of the various copolymers was well performed, kinetics of copolymerization and phase trantisition range of copolymers were widely investigated (the 6-4 copolymer seems to be the most promising but it's easy to understand that introduction of the DHPMA represents a straightforward method to control the Tcp). Explanation for the observation of the phase transition temperatures was furnished. Rational behind the work was clear. Conclusions are consistent with data. This work fills a gap in the literature about the vast area of smart copolymers (more precisely: thermo-responsive). Some issues are present: the weakest point of the work is the absence of the supplementary materials file which is mentioned in the text.

For these reasons, my overall recommendation is to reconsider after major revisions, as supplementary materials file is not visible and can not be reviewed.

/// COMMENTS///

Lines 33-34: Why past-tense (the first two verbs) when we talk about a scientific fact which happens when some conditions are met?

Figure 1d: Have you tried to optimize noise correction or concentration of copolymer to make cross-peak (3.40 ppm, 3.70 ppm) more apparent?

Figure 1e. In the footnotes of the superposition spectrum, indicate which kind of DEPT 13C spectrum has been added (135°?)

Lines 20-23: “the reactivity ratios of DHPMA and NIPAm were 20 found to be 3.09 and 0.11, suggesting that DHPMA had more preference for homopolymerization than for copolymerization while NIPAm has more preference for copolymerization than for homopolymerization.” Anyway, later in the manuscript, at line 117, the conditions for the copolymerizations are reported. If the initial concentration of co-monomers are identical, how is it possible to obtain compolymers with an excess of N-isoproylacrylamide fragments?

Lines 235-237. This phrase should be added in another section.

Why didn’t you use 13C NMR to assess the composition of the copolymers and instead this assessment was based on the use of elemental analysis?

Supplementary Materials file is not present but tables of this file are mentioned in the manuscript

Author Response

Please see the attchment.

Round 2

Reviewer 1 Report

Comments and Suggestions for Authors

The author has been revised based on my comments.

Reviewer 2 Report

Comments and Suggestions for Authors

I thank the authors for incorporating my comments into the article. In its current form, I recommend the article for publication.

Reviewer 3 Report

Comments and Suggestions for Authors

The authors of the manuscript entitled "Free radical copolymerization of N-isopropylacrylamide and 2,3-dihydroxypropyl methacrylate: Reaction kinetics and characterizations" have submitted a revised version of their work, in which they have answered point by point to the questions raised by reading the manuscript. They have improved the quality of the article and now it satisfies the requirements for its publication on Materials

My overall recommendation is to "Accept in present form"